# An Amino Acid Mixture to Counteract Skeletal Muscle Atrophy: Impact on Mitochondrial Bioenergetics

**DOI:** 10.3390/ijms25116056

**Published:** 2024-05-31

**Authors:** Francesco Bellanti, Aurelio Lo Buglio, Giuseppe Pannone, Maria Carmela Pedicillo, Ilenia Sara De Stefano, Angela Pignataro, Cristiano Capurso, Gianluigi Vendemiale

**Affiliations:** 1Department of Medical and Surgical Sciences, University of Foggia, 71122 Foggia, Italy; francesco.bellanti@unifg.it (F.B.); aurelio.lobuglio@unifg.it (A.L.B.); cristiano.capurso@unifg.it (C.C.); 2Department of Clinical and Experimental Medicine, University of Foggia, 71122 Foggia, Italy; giuseppe.pannone@unifg.it (G.P.); mariacarmela.pedicillo@unifg.it (M.C.P.); ilenia.destefano@unifg.it (I.S.D.S.); angela.pignataro@unifg.it (A.P.)

**Keywords:** immobilization, cardiotoxin, sarcopenia, amino acids

## Abstract

Skeletal muscle atrophy (SMA) is caused by a rise in muscle breakdown and a decline in protein synthesis, with a consequent loss of mass and function. This study characterized the effect of an amino acid mixture (AA) in models of SMA, focusing on mitochondria. C57/Bl6 mice underwent immobilization of one hindlimb (I) or cardiotoxin-induced muscle injury (C) and were compared with controls (CTRL). Mice were then administered AA in drinking water for 10 days and compared to a placebo group. With respect to CTRL, I and C reduced running time and distance, along with grip strength; however, the reduction was prevented by AA. Tibialis anterior (TA) muscles were used for histology and mitochondria isolation. I and C resulted in TA atrophy, characterized by a reduction in both wet weight and TA/body weight ratio and smaller myofibers than those of CTRL. Interestingly, these alterations were lightly observed in mice treated with AA. The mitochondrial yield from the TA of I and C mice was lower than that of CTRL but not in AA-treated mice. AA also preserved mitochondrial bioenergetics in TA muscle from I and C mice. To conclude, this study demonstrates that AA prevents loss of muscle mass and function in SMA by protecting mitochondria.

## 1. Introduction

Skeletal muscle represents the most important reservoir of proteins in humans and most animals, and maintenance of its integrity is important not only for preserving body structure and mobility but also for regulating glucose, lipid and protein metabolism [1]. Thus, metabolic changes that occur in skeletal muscle may influence the course of diseases so that preservation of muscle mass and function is determinant for maintaining a high quality of life [2]. Skeletal muscle atrophy can be described as a deficiency of muscle mass dependent on disuse (immobilization or bed rest) or metabolic alterations triggered by inflammation/oxidative stress [3]. Skeletal muscle atrophy and loss of mass/function rapidly occur after acute immobilization or direct injury [4,5]. Even though exercise training has been demonstrated as the most powerful intervention to preserve skeletal muscle mass and function, this strategy is feasible with difficulty in several conditions, including critical illnesses, environmental restrictions, and physical exhaustion [6,7].

To date, several amino acid-based treatments have been tried to counteract skeletal muscle atrophy, leading to controversial results [8,9,10]. However, the potential efficacy of amino acid supplementations may depend on their formulation, which should contain a mixture of both essential (EAAs) and non-essential amino acids (NEAAs, including glutamine, cysteine, and tyrosine). Indeed, whereas EAAs are more effective than NEEAs to induce net protein synthesis, glutamine reduces catabolism and modulates immune cells, cysteine mitigates oxidative stress through its thiol group, and tyrosine enhances physical performance by stimulating catecholamine neurotransmitter synthesis [11,12,13,14]. An amino acid mixture of EAAs combined with glutamine, cysteine and tyrosine demonstrated effectiveness in improving the cognitive, functional, nutritional, and clinical status of old hospitalized patients [15]. Testing this mixture on old patients with hospital-related bed immobilization, we reported its beneficial impact on skeletal muscle mass and function, as well as circulating markers of inflammation and oxidative stress [16]. Nevertheless, we could not further characterize the mechanisms underlying such positive effects in humans.

Preservation of skeletal muscle mass and function mostly relies on mitochondrial bioenergetics [17,18]. Skeletal muscle disuse induces severe mitochondrial perturbations that contribute to tissue atrophy [19]. Furthermore, mitochondrial dysfunction is pivotal in skeletal muscle atrophy induced by inflammation/oxidative stress [20]. Thus, therapeutic approaches that target mitochondria could be promising for counteracting skeletal muscle atrophy and preventing the loss of mass and function [18].

Considering these premises, we aimed to characterize the effect of an amino acid mixture (AA) containing EEAs plus glutamine, cysteine, and tyrosine in rodent models of skeletal muscle atrophy, focusing on mitochondrial bioenergetics. To this end, AA was tested in acute models of immobilization and acute inflammation/oxidative stress-induced muscle atrophy.

## 2. Results

### 2.1. Effect of AA Supplementation on Skeletal Muscle Mass and Performance

After 10 days of treatment, no modifications of body weight were reported in all the groups studied (Figure 1a). Of interest, an effect of the model (F_2,24_ = 7.961, *p* = 0.0022), treatment (F_1,24_ = 62.07, *p* < 0.0001) and interaction (F_2,24_ = 27.80, *p* < 0.0001) on TA weight was observed. In particular, with respect to control animals, muscle weight was significantly reduced in the immobilization and cardiotoxin groups treated with the placebo but not in mice treated with AA (Figure 1b). Similarly, a model (F_2,24_ = 4.648, *p* = 0.0192), treatment (F_1,24_ = 66.22, *p* < 0.0001) and interaction (F_2,24_ = 19.42, *p* < 0.0001) effect on TA muscle weight relative to body weight was reported, describing a decrease in the placebo-treated immobilization and cardiotoxin groups but not in AA-treated animals (Figure 1c). Finally, an effect of the model (F_2,24_ = 4.906, *p* = 0.0164), treatment (F_1,24_ = 9.791, *p* = 0.0046) and interaction (F_2,24_ = 4.724, *p* = 0.0186) on TA protein content was also observed. The post hoc analysis showed that the reduction in muscle protein content described after immobilization or cardiotoxin treatment, as compared to the controls, was counteracted by AA supplementation (Figure 1d).

To verify whether AA could improve skeletal muscle performance after hindlimb immobilization or cardiotoxin-induced injury, mice were subjected to a treadmill test and grip strength test at the end of the treatment period.

A significant effect of the model (F_2,24_ = 6.311, *p* = 0.0063), treatment (F_1,24_ = 12.62, *p* = 0.0016) and interaction (F_2,24_ = 4.900, *p* = 0.0164) on running time was observed. In particular, with respect to control animals, running time was significantly reduced in the immobilization and cardiotoxin groups treated with the placebo but not in mice treated with AA (Figure 2a). Correspondingly, a model (F_2,24_ = 23.12, *p* < 0.0001), treatment (F_1,24_ = 20.17, *p* = 0.0002) and interaction (F_2,24_ = 6.030, *p* = 0.0076) effect on running distance was reported, for which a decrease in the placebo-treated immobilization and cardiotoxin groups but not in AA-treated animals was described (Figure 2b). Of interest, an effect of the model (F_2,24_ = 20.87, *p* < 0.0001) and treatment (F_1,24_ = 24.09, *p* < 0.0001) on the grip strength score of the right hindlimb was observed. The post hoc analysis showed that AA supplementation counteracted the reduction in grip strength described after immobilization or cardiotoxin treatment, as compared to the controls (Figure 2c).

### 2.2. Effect of AA Supplementation on Skeletal Muscle Injury, Architecture and Regeneration

To estimate skeletal muscle injury, the activity of LDH and CPK was measured in the sera of mice after 10 days of treatment. For both enzymes’ activity, a model (F_2,24_ = 70.86, *p* < 0.0001; F_2,24_ = 25.80, *p* < 0.0001, respectively), treatment (F_1,24_ = 24.60, *p* < 0.0001; F_1,24_ = 58.26, *p* < 0.0001, respectively) and interaction (F_2,24_ = 9.104, *p* = 0.0011; F_2,24_ = 14.56, *p* < 0.0001, respectively) effect was reported. As compared to the controls, an increase in the placebo-treated immobilization and cardiotoxin groups but not in AA-treated animals was described (Figure 3a,b).

Muscle architecture was analyzed by hematoxylin and eosin staining, and representative images are shown in Figure 4a. An effect of both the model and treatment on cross-sectional area was observed (F_2,24_ = 13.60, *p* = 0.0001; F_1,24_ = 12.36, *p* = 0.0018, respectively), and a model (F_2,24_ = 13.09, *p* = 0.0001), treatment (F_1,24_ = 14.67, *p* = 0.0008) and interaction (F_2,24_ = 9.272, *p* = 0.001) effect on the mean fiber diameter of TA muscle was described. Both variables were reduced in the immobilization and cardiotoxin models treated with the placebo but not in those treated with AA supplementation (Figure 4b). Similar results were obtained by analyzing the fiber size distribution (Appendix A).

To investigate the effect of AA treatment on skeletal muscle regeneration after 10 days, the number of PAX7-positive cells in TA muscle was quantified by digital pathology (representative hotspot images are shown in Figure 5a). Of note, a model effect was observed (F_2,24_ = 13.66, *p* = 0.0001), with a significant decrease in both the immobilization and cardiotoxin groups, irrespective of treatment (Figure 5b).

### 2.3. Effect of AA Supplementation on Skeletal Muscle Mitochondria Content and Bioenergetics

To verify the impact of AA treatment on mitochondrial content and bioenergetics in the models of skeletal muscle atrophy, mitochondria were freshly isolated from TA muscle after 10 days of supplementation. Of note, a significant effect of the model (F_2,24_ = 13.30, *p* < 0.0001) and treatment (F_1,24_ = 26.86, *p* < 0.0001) on mitochondrial density was observed. The post hoc analysis showed a reduction in mitochondrial density after immobilization or cardiotoxin-induced atrophy, as compared to the controls; nevertheless, treatment with AA resulted in higher mitochondrial density with respect to the placebo (Figure 6a). We also described a model (F_2,24_ = 6.339, *p* = 0.0062), treatment (F_1,24_ = 30.35, *p* < 0.0001) and interaction (F_2,24_ = 6.259, *p* = 0.0065) effect on mitochondrial membrane potential, for which a decrease in the placebo-treated immobilization and cardiotoxin groups, recovered by AA-treated animals, was described (Figure 6b and Appendix A).

To deepen our study on mitochondrial bioenergetics in TA muscle, we detected the respiratory control ratio (RCR, the ratio between state 3—or ADP-dependent respiration—and state 4—or ADP-independent respiration), which represents an index of oxidative phosphorylation activity. Representative polarographic curves are shown in Appendix A. The RCR from Complex I- and Complex II-linked substrates was impacted by the model (F_2,24_ = 27.32, *p* < 0.0001; F_2,24_ = 14.45, *p* < 0.0001, respectively), treatment (F_1,24_ = 25.35, *p* < 0.0001; F_1,24_ = 25.51, *p* < 0.0001, respectively) and interaction (F_2,24_ = 17.63, *p* < 0.0001; F_2,24_ = 7.018, *p* = 0.0040, respectively). With respect to the controls, both indexes were significantly reduced in the immobilization and cardiotoxin groups treated with the placebo but not in groups treated with AA (Figure 6c,d). As shown by the data reported in Table 1 and Table 2, changes in the RCR were sustained by state 3 rather than state 4 respiration. Indeed, for both Complex I- and Complex II-linked substrates, an effect of the model (F_2,24_ = 16.02, *p* < 0.0001; F_2,24_ = 11.48, *p* = 0.0003, respectively), treatment (F_1,24_ = 7.27, *p* = 0.0126; F_1,24_ = 20.88, *p* = 0.0001, respectively) and interaction (F_2,24_ = 10.42, *p* = 0.0006; F_2,24_ = 7.785, *p* = 0.0025, respectively) was observed.

To verify whether changes in mitochondrial bioenergetics would cause modifications of ATP homeostasis in models of skeletal muscle atrophy, ATP content was measured in TA muscle and compared to the activity of Complex V (ATP synthase). Interestingly, an effect of the model (F_2,24_ = 16.43, *p* < 0.0001), treatment (F_1,24_ = 14.39, *p* = 0.0009) and interaction (F_2,24_ = 3.914, *p* = 0.0338) on ATP content was observed. In particular, with respect to the control animals, the ATP content was significantly reduced in the immobilization and cardiotoxin groups treated with the placebo but not in mice treated with AA (Figure 6e). On the contrary, no modifications were observed in Complex V (ATP synthase) activity between all the groups studied (Figure 6f).

## 3. Discussion

The present study demonstrates that a supplementation with an AA mixture containing EEAs plus glutamine, cysteine, and tyrosine enhances skeletal muscle mass, function, and architecture in rodent models of muscle atrophy. In particular, our results suggest that the beneficial effect of such AA supplementation is associated with the prevention of mitochondrial dysfunction and bioenergetics impairment in skeletal muscle.

Being one of the most dynamic tissues in the body, skeletal muscle accounts not only for movement but also for metabolism, thermogenesis, and protein homeostasis [21]. Even though the main skeletal muscle function consists of switching chemicals toward mechanical energy for posture and physical activity, this tissue is the most important determinant of basal energy metabolism and is a repository of amino acids required for the synthesis of organ-specific proteins and for maintaining glucose homeostasis during starvation [22]. Thus, the preservation of skeletal muscle homeostasis is crucial for retaining a satisfactory health status. Nevertheless, this homeostasis can be altered by several factors, including acute and chronic diseases, denervation, fasting, immobilization, and aging, which lead to skeletal muscle atrophy [23]. This condition severely affects the ability to counteract diseases and stress, with negative consequences on quality of life, a high socio-economic burden, and increased morbidity and mortality [24]. Even though physical exercise is currently the most effective therapy for skeletal muscle atrophy, training protocols cannot be applied for immobilized or acutely ill patients. Nutritional strategies based on AA supplementations may limit skeletal muscle atrophy induced by immobilization and counteract inflammation/oxidative stress [25,26]. Indeed, since skeletal muscle atrophy is sustained by metabolic changes that lead to increased protein degradation and reduced protein synthesis, AA supplementation may be beneficial for boosting protein synthesis and preserving muscle mass [27]. Several AA supplementations are available for these purposes, even though there is no consensus about the compound with the best efficacy for treating skeletal muscle atrophy [28,29]. We have already tested this AA mixture containing EEAs plus glutamine, cysteine, and tyrosine in a group of patients subjected to bed immobilization, reporting beneficial effects on muscle structure and function, as well as systemic markers of immune response and redox balance [16]. The results from this study confirm the effectiveness of such an AA mixture on skeletal muscle mass, architecture, and function in pre-clinical models.

First of all, this study confirms the detrimental impact that hindlimb immobilization and cardiotoxin injection exert on skeletal muscle. Indeed, surgical stapler hindlimb immobilization induces a reduction in TA mass, which is associated with a decreased myofiber size [30]. A similar effect is determined by cardiotoxin-induced TA muscle injury, which is the most reliable model for studying homogeneous damage to skeletal muscle due to a single belly and its mixed composition of fibers [31]. Even though cardiotoxin injection is not commonly applied as a model of skeletal muscle atrophy, it induces rapid fiber breakdown with extensive necrosis and inflammatory infiltration, reducing muscle mass and impairing muscle architecture [32]. Of note, supplementation with the AA mixture for 10 days resulted in the preservation of TA mass and architecture. Previous studies have demonstrated the beneficial effects of AA supplementation in murine models of hindlimb immobilization, even though supplementations were mostly enriched with branched-chain AA [33,34]. On the contrary, to our knowledge, no studies have been performed testing the effects of AA supplementation in models of cardiotoxin-induced muscle damage. EAAs and particularly branched-chain AAs (such as leucine, isoleucine, and valine) are generally described as anabolic compounds able (i) to trigger protein synthesis through the mammalian target of the rapamycin (mTOR) pathway and (ii) to inhibit protein catalysis through a reduction in ubiquitin–proteasome activity [35,36]. Even though our study was not specifically designed to investigate the impact of this AA mixture on protein homeostasis, future research will elucidate how this mechanism can be perturbated by such supplementation. It is worth noting that the AA mixture used in our study contains—other than branched-chain AAs—a great amount of glutamine, which also exerts a role in muscle maintenance [37]. Of interest, glutamine supplementation has already demonstrated effectiveness in relieving skeletal muscle loss in a rodent model of hindlimb immobilization [38].

More than being characterized by structural abnormalities, such as fading, lessening, and reducing mass and fibers, skeletal muscle atrophy leads to a functional deficit manifested by reduced force and exercise ability [39]. Both immobilization-induced and cardiotoxin-induced skeletal muscle injury caused a severe impairment in muscle performance, as demonstrated by the reduced TA muscle strength, and decreased running time/distance. Since the stimulating effect of AAs—particularly EAAs—on skeletal muscle mass could improve functional outcomes and physical performance [40], we tested the impact of our AA mixture on models of skeletal muscle atrophy, detecting preservation of muscle strength and endurance. The positive effect of this AA mixture can be further sustained by glutamine, which improves muscle power with anti-fatigue properties [41].

The serum enzymatic activity of lactate dehydrogenase (LDH) and creatin phosphokinase (CPK) is used as an indicator of skeletal muscle tissue injury, particularly after physical exercise [42,43]. Our data show that both LDH and CPK activities were increased in the sera of both models of skeletal muscle atrophy, demonstrating the occurrence of severe muscle damage. Indeed, an increase in serum LDH and CPK activity—especially in the immobilization model—can be interpreted as myonecrosis or a membrane defect consequent to skeletal muscle atrophy [44]. These results confirm the protective effect of AAs (especially branched-chain AAs and glutamine) in attenuating the increase in circulating levels of muscle injury biomarkers [45,46].

Skeletal muscle injury caused by immobilization or various toxins may trigger pathways of regeneration that activate a population of resident stem cells, named satellite cells, which are located in special niches and are identified by the marker PAX7 [47]. Myocellular damage perturbates the stem cell niche in skeletal muscle tissue, leading to PAX7 downregulation and promoting differentiation with consequent formation and fusion of new fibers. Of note, the data from our experiments show reduced expression of PAX7-positive cells in both models of skeletal muscle atrophy, suggesting that the available pool of satellite cells decreased after 10 days since receiving the regenerating stimulus. The AA mixture had no impact on the number of PAX7-positive cells, indicating that the effects of this treatment on skeletal muscle were not attributable to modifications of its regenerative potential.

Mitochondrial dysfunction plays a determinant role in the pathogenesis of skeletal muscle atrophy, since these organelles account for ATP production and regulation of metabolism, redox homeostasis, and apoptosis [48]. To elucidate a possible mechanism underlying the beneficial effects of AA mixture supplementation on TA architecture and performance in our models of skeletal muscle atrophy, several parameters related to mitochondria homeostasis were measured. First, our results confirm that models of TA atrophy are characterized by a reduction in mitochondrial density [49]. Furthermore, such models are characterized by impairment in oxidative phosphorylation, with a consequent reduction in ATP synthesis. Alterations in mitochondrial quantity are a direct factor that alters organelle function and bioenergetics in the induction of skeletal muscle atrophy [20]. It is also worth mentioning that mitochondrial dysfunction exerts a deep impact on the AA metabolism of atrophic skeletal muscle [50]. Of note, the AA mixture is able to restore mitochondrial quantity and bioenergetics, preventing ATP depletion in models of TA muscle atrophy. Nevertheless, the reduction in ATP in our models of skeletal muscle atrophy might also reflect an increase in pannexin-1, which forms ATP-permeable channels in the sarcolemma [51]. These results confirm that dietary supplementation with EAAs or branched-chain AAs is beneficial for mitochondrial homeostasis in skeletal muscle [52]. It is conceivable that such an AA mixture could stimulate mitochondrial bioenergetics via the mammalian site of the rapamycin (mTOR) pathway, increasing nicotinamide adenine dinucleotide levels and stimulating fatty acid oxidation [53]. Further studies are needed to expand our observations to other aspects of mitochondrial quality, including biogenesis, dynamics, mitophagy and apoptosis.

The present study has the following limitations, which must be considered for proper interpretation of our results. First of all, it was not designed to compare the effects of different AA mixture compositions but to provide a reliable mechanism sustaining the positive effects of a specific supplementation on skeletal muscle mass and function. Furthermore, in our investigation, we could not address different skeletal muscle types (i.e., fast-twitch and slow-twitch muscles), choosing to focus our analyses on tibialis anterior muscle, which is mostly a fast-contracting muscle [54]. It is worth mentioning that we analyzed mitochondrial bioenergetics by polarographic assessment, which allows the measurement of oxygen consumption rate with preserved organelle integrity. We did not assess the spectrophotometric activity of Complex I and II (which provides information on the maximal activities of the complexes), choosing to focus on Complex V enzymatic activity, since FoF1 ATPase function may control mitochondrial respiration [55]. Finally, our experiments were not planned to differentiate between subsarcolemmal and intermyofibrillar mitochondria. Future specific studies will clarify the particular effects of AA mixture supplementation on different skeletal muscle types and mitochondria subtypes.

In conclusion, this study demonstrates that an AA mixture containing EAAs and glutamine is able to prevent the loss of mass and function in rodent models of skeletal muscle atrophy, limiting tissue injury and architecture impairment. Such beneficial effects are not sustained through the stimulation of skeletal muscle regeneration from satellite cells but are the consequence of improved mitochondrial bioenergetics and ATP homeostasis. Hence, this study encourages further preclinical investigations and clinical trials to test the efficacy of this AA mixture in skeletal muscle atrophy.

## 4. Materials and Methods

### 4.1. Study Protocol

C57/Bl6 18-week-old male mice were used for this study. All animals were housed in conditions with a temperature of 22 °C ± 2 °C, humidity of 55% ± 10% and 12 h day/dark cycles. Mice were fed ad libitum with a standard diet (52% carbohydrates, 21% proteins, 4% lipids) and were randomized into six different groups (N = 5 animals per group), according to the following models and treatment with AA supplementation (Figure 7):Control + placebo.Control + AA.Immobilization + placebo.Immobilization + AA.Cardiotoxin + placebo.Cardiotoxin + AA.

**Figure 7 ijms-25-06056-f007:**
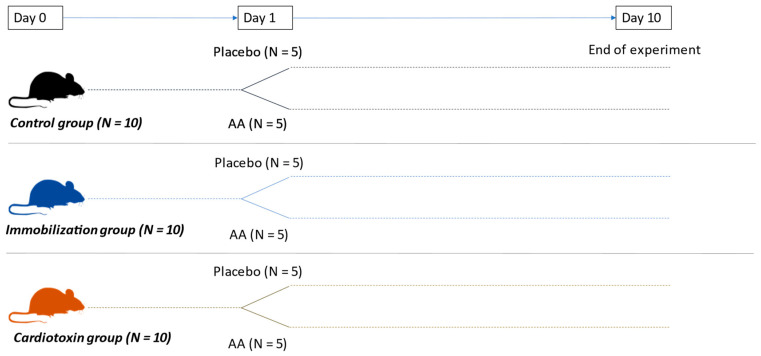
Experimental animal design.

A control set of mice was chosen—rather than contralateral hindlimb immobilization and cardiotoxin models—since the contralateral hindlimb may suffer from changes induced by the models [56]. After 10 days, muscle function was studied by both endurance running and grip strength tests. At the end of the experiment, mice were euthanized by anesthesia overdosing, and samples of blood and muscle (tibialis anterior, TA) tissue were taken. TA muscle was chosen due to its peculiar properties (single-belly skeletal muscle composed of type I, type IIa and type IIb myofibers, which are particularly rich in mitochondria, enabling its contractile function), which allow the study of both imaging and mitochondrial function [57,58]. TA samples were weighed and used for fresh mitochondria isolation.

AA supplementation was administered via oral supplementation by dissolving in a volume of water equal to 52 ± 0.3 mL (according to the estimated amount of daily consumption) at a dose of 0.1 g/kg/day for 10 days. The composition and nutritional value of AA supplementation was previously reported [16].

All animal experimental protocols were approved by the University of Foggia ethics committee and conducted in accordance with the guidelines of the Italian Ministry of Health (D.L. 26/2014) and the European Parliamentary directive 2010/63/EU.

### 4.2. Immobilization Procedure

A group of 10 C57/Bl6 mice underwent one hindlimb immobilization according to a well-established, previously published protocol [30]. Briefly, mice were anesthetized through an intraperitoneal injection of Ketamine (80 mg/kg) and Xylazine (10 mg/kg). The right hindlimb was immobilized by stapling the foot after reaching normal dorsotibial flexion through a skin stapler. One tine was inserted into the distal portion of the gastrocnemius, and the other was inserted close to the toe at the plantar portion of the foot. The hindlimb was immobilized for 10 days before removing fixing points.

### 4.3. Cardiotoxin Injection

To induce acute skeletal muscle injury-induced atrophy, a group of 10 C57/Bl6 mice underwent cardiotoxin injection in the tibialis anterior muscle following a well-defined protocol, as reported previously [59]. Briefly, mice were anesthetized through an intraperitoneal injection of Ketamine (80 mg/kg) and Xylazine (10 mg/kg). The right hindlimb was restrained, shaved with a clipper, and disinfected with 70% ethanol. Then, 10 µM cardiotoxin (217503, Merck KGaA, Darmstadt, Germany) diluted in 100 µL of phosphate buffer solution was injected into the belly of the tibialis anterior muscle.

### 4.4. Running Test

A homemade rodent treadmill was used, as previously reported [60]. The running test consisted of a running session at a speed of 9 cm/sec with an inclination of 5°. The speed was increased by 3 cm/sec every 12 min. A rigid brush, positioned in the cool-down area at the end of the treadmill track, was used to motivate the animals to perform the exercise. The test was recorded, and the video was analyzed to calculate the mice’s permanence in the cool-down zone. Mice that needed at least 5 stimuli in the cool-down zone were considered “exhausted” (with the conclusion of the exercise).

### 4.5. Muscle Strength

Skeletal muscle strength was assessed according to a previously described procedure [61]. Animals were held by their tail and allowed to grasp weights with the right hindlimb for at least 3 s (Appendix A). If the test was successful, mice were tested for the next heaviest weight. If mice dropped the weight in less than 3 s, the test was repeated after 10 s of rest. If the test failed three times, the trial was stopped, and the animal was assigned the maximum weight achieved. A final score was calculated as the sum of the weight held for 3 s plus the heaviest weight dropped before 3 s multiplied by the time it was held.

### 4.6. Serum Creatin Phosphokinase (CPK) and Lactate Dehydrogenase (LDH) Measurement

The serum was derived from the centrifugation of whole blood collected just after sacrifice. CPK activity was measured by using a diagnostic ELISA kit (MBS755902, MyBioSource, San Diego, CA, USA), according to the manufacturer procedure. LDH activity was measured by using a colorimetric kit (ab102526, Abcam, Cambridge, UK), according to the manufacturer’s instructions.

### 4.7. Histology and Immunohistochemistry

After excision, portions of TA muscle from each mouse were placed in 4% formalin for 48 h, and subsequently, paraffin was embedded, cut, deparaffinized and used for histology or immunohistochemistry staining.

Histological analysis of TA specimens was performed following hematoxylin and eosin (H&E) staining. Five randomly selected sections from the mid-belly of each TA muscle were selected for analysis. The cross-sectional area (CSA), the mean diameter of at least 70 muscle fibers (regardless of fiber types) and the fiber size distribution were determined in over 40 myofibers/field from at least 5 different fields (×20 magnification) by using dedicated software (ImageJ, version 1.48).

Immunohistochemical analysis was performed on 4 μm serial sections by using Ventana Benchmark^®^ XT autostainer (Roche Diagnostics International AD, Rotkreuz, Switzerland) and standard linked streptavidin–biotin horseradish peroxidase technique (LSAB-HRP), following the best protocol for primary rabbit polyclonal antibody anti-mouse PAX7, diluted 1:100 in PBS (MBS9202728, MyBioSource, San Diego, CA, USA) and incubated overnight. Negative control slides without primary antibodies were included for each staining. The results of the staining were independently evaluated by two operators. A total of 10 representative high-power fields for each section were analyzed with an optical microscope (Zeiss Axioscope, Carl Zeiss Microscopy, White Plains, NY, USA). Sections were digitally scanned with NanoZoomer S60 C13210 series Hamamatsu Photonics K-K (Hamamatsu Photonics, Hamamatsu City, Japan), and immunostained sections were evaluated with Visiopharm software version 2021.02 (APP tune, APP author, Deep Learning with Author AI).

### 4.8. Mitochondria Isolation and Bioenergetic Analysis

Mitochondria were isolated from TA muscle by differential centrifugation, as previously described [62]. Protein concentration was determined using the Lowry micromethod kit (TP0300, Merck KGaA, Darmstadt, Germany), and mitochondrial density was calculated and expressed as mg of protein per g of skeletal muscle tissue.

Mitochondrial membrane potential was assessed at 37 °C after incubating approximately 0.5 mg of protein/mL of mitochondria in a thermostatically equipped oxygraph+ chamber (Hansatec Instruments Ltd., Norfolk, UK) provided with a Clark’s electrode and a tetraphenylphosphonium (TPP+) electrode (WPI, Berlin, Germany) in the presence of succinate, rotenone and oligomycin. Membrane potential was calculated using a modified Nernst equation, as previously reported [62].

Oxygen uptake in state 3 and state 4 was determined using 10 mM glutamate + 5 mM malate or 2 mM succinate as Complex I-linked or Complex II-linked oxidative substrates, respectively; the respiratory control ratio (RCR) was then calculated, as previously reported [62]. The purity of the mitochondrial fraction was validated by an RCR ≥ 4 measured in the control samples [63].

ATPase activity was evaluated spectrophotometrically, measuring ATP hydrolysis with an ATP-regenerating system coupled to NADPH oxidation [62]. ATP concentration was measured in samples of TA muscle by using a bioluminescence kit (A22066, Thermofisher Scientific Inc, Waltham, MA, USA) according to the manufacturer’s instructions.

### 4.9. Statistical Analysis

Data were expressed as the mean ± standard deviation of the mean (SD). To compare all groups, we used a two-way analysis of variance (ANOVA) to test the main effect of the model (immobilization or cardiotoxin injection) and the treatment as between-subject factors; the interaction model × treatment was studied, and a Tukey test was applied as a post hoc multiple comparison test. All tests were two-sided, and *p* < 0.05 was considered statistically significant. Statistical analysis was performed with the package Graph-Pad Prism 6 for Windows (GraphPad Software Inc., San Diego, CA, USA).

## Figures and Tables

**Figure 1 ijms-25-06056-f001:**
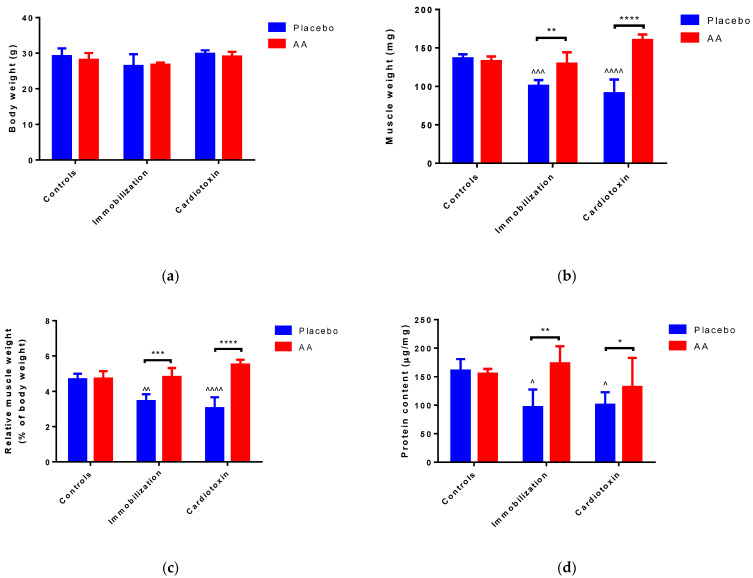
Impact of AA on skeletal muscle mass. (**a**) Mice body weight; (**b**) Tibialis anterior (TA) muscle weight; (**c**) TA weight relative to body weight; (**d**) Protein content in TA muscle. Data are expressed as mean ± standard deviation of 5 different experiments. Statistical differences were assessed by two-way analysis of variance (ANOVA) and Tukey’s post hoc test. ^ = *p* < 0.05, ^^ = *p* < 0.01, ^^^ = *p* < 0.001, ^^^^ = *p* < 0.0001 vs. control groups; * = *p* < 0.05; ** = *p* < 0.01; *** = *p* < 0.001, **** = *p* < 0.0001 vs. placebo groups.

**Figure 2 ijms-25-06056-f002:**
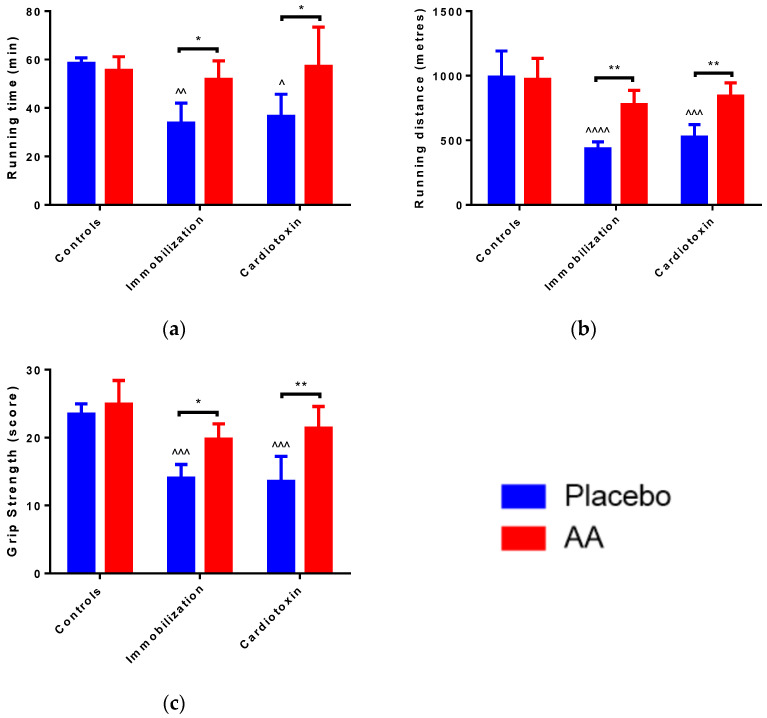
Impact of AA on skeletal muscle performance. (**a**) Running duration for treadmill test; (**b**) Running distance for treadmill test; (**c**) Hind limb force score for grip strength test. Data are expressed as mean ± standard deviation of 5 different experiments. Statistical differences were assessed by two-way analysis of variance (ANOVA) and Tukey’s post hoc test. ^ = *p* < 0.05, ^^ = *p* < 0.01, ^^^ = *p* < 0.001, ^^^^ = *p* < 0.0001 vs. control groups; * = *p* < 0.05; ** = *p* < 0.01 vs. placebo groups.

**Figure 3 ijms-25-06056-f003:**
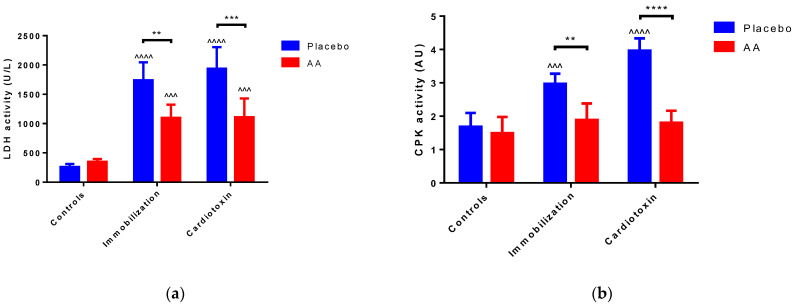
Impact of AA on skeletal muscle damage. (**a**) Serum lactate dehydrogenase (LDH) activity; (**b**) Serum creatin phosphokinase (CPK) activity. Data are expressed as mean ± standard deviation of 5 different experiments. Statistical differences were assessed by two-way analysis of variance (ANOVA) and Tukey’s post hoc test. ^^^ = *p* < 0.001, ^^^^ = *p* < 0.0001 vs. control groups; ** = *p* < 0.01, *** = *p* < 0.001, **** = *p* < 0.0001 vs. placebo groups.

**Figure 4 ijms-25-06056-f004:**
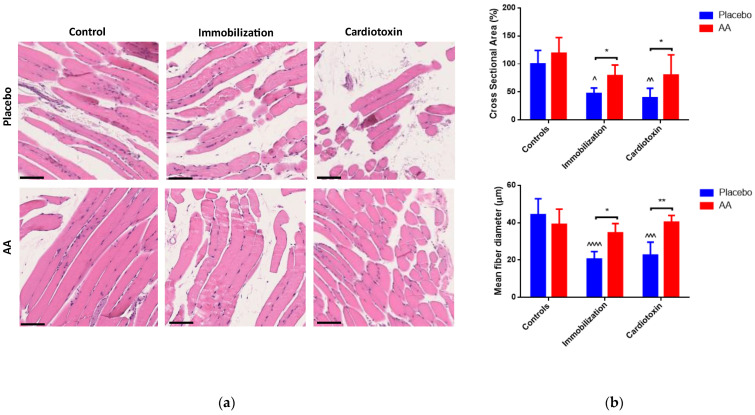
Impact of AA on skeletal muscle architecture. (**a**) Representative images of hematoxylin and eosin staining of tibialis anterior muscle (scale bars = 250 μm); (**b**) Cross-sectional area (% of controls, top panel) and mean fiber diameter (bottom panel) of tibialis anterior muscle. Data are expressed as mean ± standard deviation of 5 different experiments. Statistical differences were assessed by two-way analysis of variance (ANOVA) and Tukey’s post hoc test. ^ = *p* < 0.05, ^^ = *p* < 0.01, ^^^ = *p* < 0.001, ^^^^ = *p* < 0.0001 vs. control groups; * = *p* < 0.05, ** = *p* < 0.01 vs. placebo groups.

**Figure 5 ijms-25-06056-f005:**
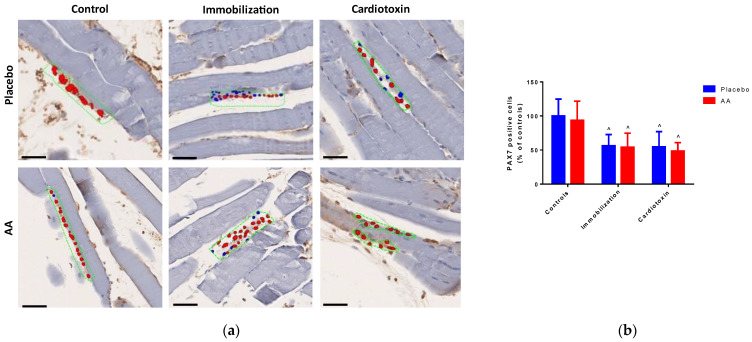
Impact of AA on skeletal muscle regeneration. (**a**) Representative images of hotspot analysis of PAX7 staining (red color) of tibialis anterior muscle (scale bars = 250 μm), blue color indicates cell nuclei, while green dashed boxes indicate positive staining areas; (**b**) PAX7-positive cells (% of controls, top panel) in tibialis anterior muscle. Data are expressed as mean ± standard deviation of 5 different experiments. Statistical differences were assessed by two-way analysis of variance (ANOVA) and Tukey’s post hoc test. ^ = *p* < 0.05 vs. control groups.

**Figure 6 ijms-25-06056-f006:**
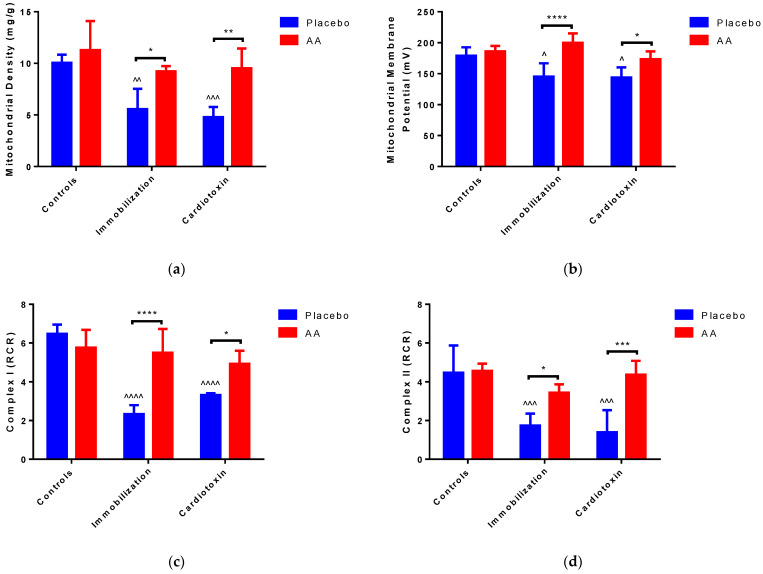
Impact of AA on skeletal muscle mitochondria density and bioenergetics. (**a**) Mitochondrial density in tibialis anterior (TA) muscle; (**b**) Membrane potential in mitochondria from TA muscle; (**c**) Respiratory control ratio (RCR) of mitochondrial Complex I from TA muscle; (**d**) RCR of mitochondrial Complex II from TA muscle; (**e**) ATP content in TA muscle; (**f**) Enzymatic activity of ATPase (Complex V) from TA muscle. Data are expressed as mean ± standard deviation of 5 different experiments. Statistical differences were assessed by two-way analysis of variance (ANOVA) and Tukey’s post hoc test. ^ = *p* < 0.05, ^^ = *p* < 0.01, ^^^ = *p* < 0.001, ^^^^ = *p* < 0.0001 vs. control groups; * = *p* < 0.05; ** = *p* < 0.01; *** = *p* < 0.001; **** = *p* < 0.0001.

**Table 1 ijms-25-06056-t001:** Oxygen consumption rates in mitochondria isolated from tibialis anterior muscle of different models used in this study, adding Complex I-linked substrates.

Glutamate + Malate	State 3 Respiration (nmol O_2_/min/mg)	State 4 Respiration (nmol O_2_/min/mg)
Controls + Placebo	10.09 ± 2.04	1.56 ± 0.21
Controls + AA	7.94 ± 2.18	1.38 ± 0.17
Immobilization + Placebo	3.49 ± 0.83 ^^^^	1.49 ± 0.19
Immobilization + AA	7.09 ± 1.44 *	1.29 ± 0.12
Cardiotoxin + Placebo	4.35 ± 1.03 ^^^	1.31 ± 0.24
Cardiotoxin + AA	7.54 ± 1.44 *	1.53 ± 0.21

Statistical differences were assessed by two-way analysis of variance (ANOVA) and Tukey’s post hoc test. ^^^ = *p* < 0.001, ^^^^ = *p* < 0.0001 vs. control groups; * = *p* < 0.05 vs. placebo groups.

**Table 2 ijms-25-06056-t002:** Oxygen consumption rates in mitochondria isolated from tibialis anterior muscle of different models used in this study, adding Complex II-linked substrates.

Succinate	State 3 Respiration (nmol O_2_/min/mg)	State 4 Respiration (nmol O_2_/min/mg)
Controls + Placebo	19.27 ± 4.88	4.31 ± 1.15
Controls + AA	18.15 ± 4.25	3.98 ± 1.32
Immobilization + Placebo	7.20 ± 2.79 ^^^	4.14 ± 0.99
Immobilization + AA	15.51 ± 4.07 *	4.51 ± 1.91
Cardiotoxin + Placebo	5.89 ± 2.15 ^^^	4.24 ± 0.94
Cardiotoxin + AA	17.87 ± 4.14 ***	4.09 ± 1.56

Statistical differences were assessed by two-way analysis of variance (ANOVA) and Tukey’s post hoc test. ^^^ = *p* < 0.001 vs. control groups; * = *p* < 0.05, *** = *p* < 0.001 vs. placebo groups.

## Data Availability

The authors confirm that the data supporting the findings of this study are available within the article. The raw data that support the findings of this study are available from the corresponding author upon reasonable request.

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
