# Peer review of "An Amino Acid Mixture to Counteract Skeletal Muscle Atrophy: Impact on Mitochondrial Bioenergetics"

_ijms, 2024, doi:10.3390/ijms25116056_

Round 1

Reviewer 1 Report

Comments and Suggestions for Authors

This study describes the effect of an amino acid mixture in two animal models of skeletal muscle atrophy. Mice were subjected to immobilization of one hindlimb or cardiotoxin-induced muscle injury. Then, they were administered the amino acid mixture in drinking water. Both models presented reduced muscle functions and the reduction was prevented by the amino acid mixture. In both models the tibialis anterior muscles resulted in atrophy and the alterations were less pronounced in mice treated with the amino acid mixture. Treatment with the amino acid mixture partially preserved the mitochondrial bioenergetics in muscles from both animal models. Most of the data is significant and clearly presented, but some issued could be improved.

Minor issues

Line 49; Please indicate which NT is affected.

In Figure 2,3 and 4 it would be important to include the statistical difference of result obtained after AA-treatment compared to control conditions.

Figure 4; very difficult to evaluate the cross sectional area of myofibers if most fibers seem to be cut in tangential or longitudinal way.

Figure legend 5; Please indicate which color revels Pax2 positive cells.

It has been reported that disuse induce a significant increase in pannexin1 that form ATP permeable channels in the cell membrane (https://doi.org/10.3390/biology12030431). Please discus your findings on ATP levels keeping in mind this report because the reduction in ATP might not reflect only a mitochondrial dysfunction. This paper could also be mentioned in the introduction.

In the discussion section, line 180, please rephase it because the AA mixture prevents the changes observed in untreated animals. Line 180; there is no improvement since it prevents. Line 239; There is no recovery since treatment was applied simultaneous to the atrophy-inducing condition.

Author Response

This study describes the effect of an amino acid mixture in two animal models of skeletal muscle atrophy. Mice were subjected to immobilization of one hindlimb or cardiotoxin-induced muscle injury. Then, they were administered the amino acid mixture in drinking water. Both models presented reduced muscle functions and the reduction was prevented by the amino acid mixture. In both models the tibialis anterior muscles resulted in atrophy and the alterations were less pronounced in mice treated with the amino acid mixture. Treatment with the amino acid mixture partially preserved the mitochondrial bioenergetics in muscles from both animal models. Most of the data is significant and clearly presented, but some issued could be improved.

Reply: we thank the reviewer for his positive comments. We hope that the revised version of the manuscript is improved after addressing his issues and the concerns of other reviewers.

Minor issues

Line 49; Please indicate which NT is affected.

Reply: we specified “catecholamine neurotransmitter synthesis” (line 46).

In Figure 2,3 and 4 it would be important to include the statistical difference of result obtained after AA-treatment compared to control conditions.

Reply: according to the reviewer’s request, we included the statistical difference of results between AA-treatment vs controls, which was significant only for LDH activity (figure 3).

Figure 4; very difficult to evaluate the cross sectional area of myofibers if most fibers seem to be cut in tangential or longitudinal way.

Reply: we replaced the figure 4 including new representative images of fibers.

Figure legend 5; Please indicate which color revels Pax2 positive cells.

Reply: Pax7 positive cells are revealed by the red color. This information is now added in the figure 5 legend, as requested.

It has been reported that disuse induce a significant increase in pannexin1 that form ATP permeable channels in the cell membrane (https://doi.org/10.3390/biology12030431). Please discus your findings on ATP levels keeping in mind this report because the reduction in ATP might not reflect only a mitochondrial dysfunction. This paper could also be mentioned in the introduction.

Reply: we thank the reviewer for his suggestion. We included this report and mentioned the paper in the discussion section (lines 292-294).

In the discussion section, line 180, please rephase it because the AA mixture prevents the changes observed in untreated animals. Line 180; there is no improvement since it prevents. Line 239; There is no recovery since treatment was applied simultaneous to the atrophy-inducing condition.

Reply: we agree with the comments of the reviewer. Discussion was rephrased at lines 197-198 (“…prevention of mitochondrial dysfunction and bioenergetics impairment…” rather than “improvement”) and line 256 (“…detecting a preservation of…” rather than “recovery”).

Reviewer 2 Report

Comments and Suggestions for Authors

This is a classic study demonstrating the positive effects of amino acid mixture supplementation on skeletal muscle atrophy. This approach has worked well both in models of induced muscle lesions and in models of hereditary muscle pathologies. For this reason, I have questions about the novelty of the study, as well as a number of additional important comments.

Comments:

1. The main question for the discussion section. I still don't understand the mechanism of action of AA supplementation, especially how it affects mitochondria. All mitochondrial indicators that the authors show are the result of the action of AA supplementation, but do not show the mechanism in any way. Authors must provide a convincing description of the possible effects of AA supplementation.

2. Justify the choice of tibialis anterior muscle in experiments. This is a small enough muscle to be used for histological analysis and evaluation of so many mitochondrial parameters (mitochondrial volume must be very small).

3. Respiratory control ratio classically shows the coupling of respiration and phosphorylation, it is better to indicate this. In this case, it is extremely important to see what is causing the decrease in RCR (decrease in state 3 or increase in state 4?). Present these data in the form of a table or graphs, as well as the original polarographic curves (the latter can be shown in the supplementary materials).

Justify the choice of such low concentrations of glutamate/malate and succinate, it is much less than physiological.

4. The authors measured the activity of complex V only. What is the reason for this choice? It is known that muscle pathologies and muscle lesions lead to changes in the level of other ETC complexes (often complex I), why this choice was made is not yet clear.  

5. In supplementary materials, it is necessary to present original curves demonstrating changes in the membrane potential of mitochondria.

6. Regarding Fig. 4, it is better to present not only the average diameters, but also show the fiber size distribution.

7. Additional figure must be presented in the main text.

Author Response

This is a classic study demonstrating the positive effects of amino acid mixture supplementation on skeletal muscle atrophy. This approach has worked well both in models of induced muscle lesions and in models of hereditary muscle pathologies. For this reason, I have questions about the novelty of the study, as well as a number of additional important comments.

Comments:

  1. The main question for the discussion section. I still don't understand the mechanism of action of AA supplementation, especially how it affects mitochondria. All mitochondrial indicators that the authors show are the result of the action of AA supplementation, but do not show the mechanism in any way. Authors must provide a convincing description of the possible effects of AA supplementation.

Reply: we thank the reviewer for his positive comments. We agree with him about the mechanistical limitation of our study. To comply with his request, we discussed the possible mitochondrial mechanism of AA supplementation. Indeed, it is conceivable that the AA mixture could stimulate mitochondrial bioenergetics via the mammalian site of the rapamycin (mTOR) pathway, increasing nicotinamide adenine dinucleotide levels and stimulating fatty acid oxidation. This is now stated in the revised manuscript (discussion section, lines 295-298).

  1. Justify the choice of tibialis anterior muscle in experiments. This is a small enough muscle to be used for histological analysis and evaluation of so many mitochondrial parameters (mitochondrial volume must be very small).

Reply: tibialis anterior muscle is a single belly skeletal muscle composed of type I, type IIa and type IIb myofibers, particularly rich in mitochondria to enable its contractile function. Due to its peculiar properties, TA muscle is a suitable model for studying both imaging and mitochondrial function (PMC5226003, PMC7853988). We added this information in the revised version of the manuscript (lines 343-345).

  1. Respiratory control ratio classically shows the coupling of respiration and phosphorylation, it is better to indicate this. In this case, it is extremely important to see what is causing the decrease in RCR (decrease in state 3 or increase in state 4?). Present these data in the form of a table or graphs, as well as the original polarographic curves (the latter can be shown in the supplementary materials).

Justify the choice of such low concentrations of glutamate/malate and succinate, it is much less than physiological.

Reply: the decrease in RCR was dependent on a decreased state 3 rather than increased state 4. To comply with the reviewer’s request, we added these data in tables 1 and 2, and representative polarographic curves were presented in the supplementary materials (Figures S3-S4). A description of such results was also added in the main text (lines 157-158, lines 163-167). The concentrations of glutamate/malate and succinate were erroneously reported as micromoles but are now corrected as millimoles (lines 422-423).

  1. The authors measured the activity of complex V only. What is the reason for this choice? It is known that muscle pathologies and muscle lesions lead to changes in the level of other ETC complexes (often complex I), why this choice was made is not yet clear.

Reply: we thank the reviewer for this observation. We analyzed mitochondrial bioenergetics by polarographic assessment, which allows measurement of oxygen consumption rate with preserved organelle integrity. Mitochondrial respiration was assessed by using both Complex I- and Complex II-linked substrates. Since we found remarkable differences in RCR – relying on State 3, or ADP-dependent respiration – in both conditions, we decided not to measure spectrophotometric activity of Complex I and II (which provides information on the maximal activities of the complexes). Consistent with impaired ADP-dependent respiration in skeletal muscle atrophy models, we reported low tissue ATP content. Thus, we chose to focus on Complex V enzymatic activity, since FoF1 ATPase function may control mitochondrial respiration (PMID 20226160). Nevertheless, our choice was mentioned in the discussion section, as a potential limitation of this study (lines 307-312).

  1. In supplementary materials, it is necessary to present original curves demonstrating changes in the membrane potential of mitochondria.

Reply: accordingly, we added a supplementary figure (S2) representing changes in mitochondrial membrane potential.

  1. Regarding Fig. 4, it is better to present not only the average diameters, but also show the fiber size distribution.

Reply: the fiber size distribution is now represented in a new figure (S1). The main text was also modified in the results (lines 124-125) and methods sections (lines 398-399).

  1. Additional figure must be presented in the main text.

Reply: accordingly, the figure S1 is now included in the main text as figure 5.

Reviewer 3 Report

Comments and Suggestions for Authors

Francesco Bellanti and colleagues investigated the effects of amino acid mixture supplementation on skeletal muscle atrophy, which is induced by immobilization or cardiotoxin injection. The authors measured parameters associated with mouse physical activity, muscle injury, mitochondria functions, etc. The authors concluded that amino acid mixture supplementation prevented muscle atrophy and promoted muscle function. It appeared that supplementation of amino acid mixtures protects muscle atrophy. However, the experimental design was not well organized. Furthermore, the authors did not explain why the amino acid mixture protects muscle atrophy.

Experimental design

The authors divided 3 groups to assess the effects of amino acid supplements on muscle atrophy: control, immobilization, and cardiotoxin. Why did the authors NOT use the contralateral hindlimb as an immobilization or a cardiotoxin injection control?

Cardiotoxin injection

The authors took advantage of cardiotoxin injection as one of the muscle atrophy models. However, the cardiotoxin injection model is not utilized as a muscle atrophy model compared to the muscle immobilization model. What is the significance of using the cardiotoxin injury model as a muscle mass reduction model?

Figure 2

The authors measured grip strength as shown in Fig. 2c. Why was grip strength reduced in immobilized and cardiotoxin-injected mice? In these experiments, hind limbs might be atrophied or injured, but forelimbs should be intact.

Figure 3

Why were LDH activity and CPK activity increased in the immobilization group? What did these results mean?

Figure 4

HE sections were so poor that the details of muscles could not be observed. Were these muscle cross-sections? Did the authors measure the cross-sectional area or fiber diameter using these pictures? If so, the authors should make muscle cross-sections again because these sections were poor to analyze.

Figure 5

The distribution of Pax7-positive cells was not convinced. Pax7-positive cells should be localized between myofibers and the basement membrane in the control muscles.

Figure 6

The authors prepared mitochondria fraction but did not show the purity of the mitochondria fraction. How did the authors guarantee the purity of their mitochondria fraction?

Comments on the Quality of English Language

Minor editing of English language required.

Author Response

Francesco Bellanti and colleagues investigated the effects of amino acid mixture supplementation on skeletal muscle atrophy, which is induced by immobilization or cardiotoxin injection. The authors measured parameters associated with mouse physical activity, muscle injury, mitochondria functions, etc. The authors concluded that amino acid mixture supplementation prevented muscle atrophy and promoted muscle function. It appeared that supplementation of amino acid mixtures protects muscle atrophy. However, the experimental design was not well organized. Furthermore, the authors did not explain why the amino acid mixture protects muscle atrophy.

Reply: we understand the main criticisms of the reviewer. We hope that the modifications applied to the manuscript in order to address his concerns and the concerns of other reviewers are satisfactory.

Experimental design

The authors divided 3 groups to assess the effects of amino acid supplements on muscle atrophy: control, immobilization, and cardiotoxin. Why did the authors NOT use the contralateral hindlimb as an immobilization or a cardiotoxin injection control?

Reply: we chose not to use the contralateral hindlimb as a control since it can be subjected to alterations induced by the model, as previously reported (PMID 10614762). This is now stated in the method section (lines 338-340).

Cardiotoxin injection

The authors took advantage of cardiotoxin injection as one of the muscle atrophy models. However, the cardiotoxin injection model is not utilized as a muscle atrophy model compared to the muscle immobilization model. What is the significance of using the cardiotoxin injury model as a muscle mass reduction model?

Reply: we thank the reviewer for his observation. We aimed not to limit our experiments to test the effect of the AA mixture in immobilization-induced atrophy. Thus, we applied the cardiotoxin injection model since it causes a rapid breakdown of skeletal muscle fibers, as well as extensive necrosis and inflammatory infiltration, simulating injury-induced atrophy (PMID: 26807982). We added this comment in the discussion section (lines 230-233). It is worth noting that complete muscle regeneration after a single cardiotoxin injection occurs in 1 month, while our experiments were performed after 10 days of AA mixture administration.

Figure 2

The authors measured grip strength as shown in Fig. 2c. Why was grip strength reduced in immobilized and cardiotoxin-injected mice? In these experiments, hind limbs might be atrophied or injured, but forelimbs should be intact.

Reply: as described in the main text (lines 380-381), grip strength was assessed on hindlimbs (and not forelimbs). To further represent the procedure, figure S5 was added in supplementary materials.

Figure 3

Why were LDH activity and CPK activity increased in the immobilization group? What did these results mean?

Reply: an increase in serum LDH and CPK activity in the immobilization group can be interpreted as myonecrosis or membrane defect consequent to skeletal muscle atrophy (PMID: 22137979). This point has been added in the discussion section of the revised manuscript (lines 263-265).

Figure 4

HE sections were so poor that the details of muscles could not be observed. Were these muscle cross-sections? Did the authors measure the cross-sectional area or fiber diameter using these pictures? If so, the authors should make muscle cross-sections again because these sections were poor to analyze.

Reply: Figure 4 was replaced with new representative images also to comply with the request of reviewer 1.

Figure 5

The distribution of Pax7-positive cells was not convinced. Pax7-positive cells should be localized between myofibers and the basement membrane in the control muscles.

Reply: Figure 5 was replaced with representative images showing the localization of Pax7-positive cells between myofibers and the basement membrane.

Figure 6

The authors prepared mitochondria fraction but did not show the purity of the mitochondria fraction. How did the authors guarantee the purity of their mitochondria fraction?

Reply: even though we did not detect contamination of mitochondrial isolates by western blot, we obtained a respiratory control index ≥4, which is considered as evidence of a viable mitochondria preparation (PMID 21490576). We added this statement in the method section of the revised manuscript (lines 425-426).

Round 2

Reviewer 2 Report

Comments and Suggestions for Authors

The authors need to make further corrections.

1. The authors presented polarographic curves. However, it is not clear why the authors presented only part of the curve; why do the authors not present the curve after the ADP is exhausted? This seems illogical. I also recommend that authors double-check their data. I may be wrong, but the data presented in Tables 1 and 2 appears to be an underestimate. What protein concentration was used in the experiments?

2.  Figure S2. Why are the curves presented called polarographic? The potential assessment in this case is based on the use of a TPP-sensitive electrode. I recommend that the authors describe the measurement methodology in more detail. As presented, these curves do not provide any information.

3. Fig. S1. The authors gave the fiber diameters in mcm. However, the presented figures are discouraging; the diameter of mouse fibers cannot reach 3000 mcm!

4. Photo S5 is of very poor quality. Please improve.

5. How was the area selected for histological analysis? Was this choice objective? Is 70 fibers enough for analysis? This seems questionable given that TA contains significantly more fiber. It would be more correct to evaluate all fibers in the muscle.

Author Response

The authors need to make further corrections.

  1. The authors presented polarographic curves. However, it is not clear why the authors presented only part of the curve; why do the authors not present the curve after the ADP is exhausted? This seems illogical. I also recommend that authors double-check their data. I may be wrong, but the data presented in Tables 1 and 2 appears to be an underestimate. What protein concentration was used in the experiments?

Reply: we replaced Figures S3 and S4 presenting all the polarographic curves. We also double-checked our data, and we confirm results presented in Tables 1 and 2. Probably the Reviewer refers to nAtoms O/min/mg, but we expressed our data as nmolO2/min/mg. We used approximately a protein concentration of 0.5 mg/ml in our experiments. This information is now reported in the method section (lines 418-419).

  1. 2.  Figure S2. Why are the curves presented called polarographic? The potential assessment in this case is based on the use of a TPP-sensitive electrode. I recommend that the authors describe the measurement methodology in more detail. As presented, these curves do not provide any information.

Reply: we modified the Figure S2 legend, describing the methodology more in detail, as requested.

  1. Fig. S1. The authors gave the fiber diameters in mcm. However, the presented figures are discouraging; the diameter of mouse fibers cannot reach 3000 mcm!

Reply: we mistakenly gave the fiber diameters in μM rather than μM2. The figure S1 legend was corrected.

  1. Photo S5 is of very poor quality. Please improve.

Reply: we replaced photo S5 with a less blurred photo.

  1. How was the area selected for histological analysis? Was this choice objective? Is 70 fibers enough for analysis? This seems questionable given that TA contains significantly more fiber. It would be more correct to evaluate all fibers in the muscle.

Reply: the number of 70 fibers was considered as the minimum threshold for analysis. However, histological analysis was performed on over 40 myofibers/field from at least five different fields (Ñ…20 magnification). This is now stated in the method section (lines 399-400). 

Round 3

Reviewer 2 Report

Comments and Suggestions for Authors

1. Question regarding the curves in Fig. S3. Show both scales in the same range. If all ranges are the same (the inscriptions are not visible, this is also a drawback of the figure), then, for example, on the control + AA curve, the respiration rate in state 4 seems to be significantly higher compared to the control group.

2. In Fig. 4, the authors gave the fiber size in mcm. While in Fig. S1 size is given in µM2, firstly, this abbreviation means micromoles, and secondly, the value is given in square. What does all of this mean? It's discouraging.

3. I still don't understand the choice of fibers used to calculate their sizes. For objectivity, the authors should evaluate the fibers throughout the entire section; this is the only correct approach. And it is important to clarify how the muscle region used for assessment was selected. The question is the same, was this an objective analysis? This is due to the fact that the TA is a fairly long muscle and the measurement results may depend on the area taken for analysis.  

4. Photo S5 has not changed. Do I understand correctly that muscle strength was assessed only for one (right) hind limb?

Author Response

  1. Question regarding the curves in Fig. S3. Show both scales in the same range. If all ranges are the same (the inscriptions are not visible, this is also a drawback of the figure), then, for example, on the control + AA curve, the respiration rate in state 4 seems to be significantly higher compared to the control group.

Reply: to comply with the reviewer’s comment, we replaced figures S3 and S4 using scales of both x and y axes in the same range. For better clarity, we deleted the grids. We apologize but we cannot magnify the inscriptions.

  1. In Fig. 4, the authors gave the fiber size in mcm. While in Fig. S1 size is given in µM2, firstly, this abbreviation means micromoles, and secondly, the value is given in square. What does all of this mean? It's discouraging.

Reply: there was a mistake in replacing figure S1 during the past revision. The scale of x axis is now corrected.

  1. I still don't understand the choice of fibers used to calculate their sizes. For objectivity, the authors should evaluate the fibers throughout the entire section; this is the only correct approach. And it is important to clarify how the muscle region used for assessment was selected. The question is the same, was this an objective analysis? This is due to the fact that the TA is a fairly long muscle and the measurement results may depend on the area taken for analysis.  

Reply: to render the analysis as much objective as possible, fibers were evaluated by assessing the mid-belly sections of TA muscle region. This information has been added to the method section (lines 398-399).

  1. Photo S5 has not changed. Do I understand correctly that muscle strength was assessed only for one (right) hind limb?

Reply: we got a single picture of grip strength test during our experiments, which is blurred and which we tried to sharpen with the help of a photo software. The reviewer was correct in understanding that muscle strength was assessed only for one hindlimb.

Round 4

Reviewer 2 Report

Comments and Suggestions for Authors

The authors have improved the manuscript sufficiently